# Self-Evolving Pseudo-Rehearsal for Catastrophic Forgetting with Task Similarity in LLMs

**Jun Wang**[1][*] **Liang Ding**[2][*] **Shuai Wang**[1] **Hongyu Li**[3] **Yong Luo**[1][†] **Huangxuan Zhao**[1][†]
**Han Hu**[4] **Bo Du**[1]

[1]School of Computer Science, National Engineering Research Center of Multimedia Software
and Hubei Key Laboratory of Multimedia and Network Communication Engineering,
Wuhan University, Wuhan, China
[2]The University of Sydney, Australia   [3]Individual Researcher
[4]School of Information and Electronics, Beijing Institute of Technology, Beijing, China
`{junwang_ai, wangshuai123, luoyong, zhaohuangxuan, dubo}@whu.edu.cn`
`{liangding.liam, hongyuli102799}@gmail.com   hhu@bit.edu.cn`

## Abstract

Continual learning for large language models (LLMs) demands a precise balance between **plasticity** - the ability to absorb new tasks - and **stability** - the preservation of previously learned knowledge. Conventional rehearsal methods, which replay stored examples, are limited by long-term data inaccessibility; earlier pseudo-rehearsal methods require additional generation modules, while self-synthesis approaches often generate samples that poorly align with real tasks, suffer from unstable outputs, and ignore task relationships. We present *Self-Evolving Pseudo-Rehearsal for Catastrophic Forgetting with Task Similarity* (**SERS**), a lightweight framework that 1) decouples pseudo-input synthesis from label creation, using semantic masking and template guidance to produce diverse, task-relevant prompts without extra modules; 2) applies label self-evolution, blending base-model priors with fine-tuned outputs to prevent over-specialization; and 3) introduces a dynamic regularizer driven by the Wasserstein distance between task distributions, automatically relaxing or strengthening constraints in proportion to task similarity. Experiments across diverse tasks on different LLMs show that our SERS reduces forgetting by over 2% points against strong pseudo-rehearsal baselines, by ensuring efficient data utilization and wisely transferring knowledge. The code will be released at `https://github.com/JerryWangJun/LLM_CL_SERS`.

## 1 Introduction

Enabling large language models (LLMs) to acquire new knowledge continuously (Wu et al., 2024; Zheng et al., 2025b) holds significant importance for developing artificial intelligence systems with lifelong learning abilities. While practical applications demand LLMs continually adapt to evolving downstream tasks, conventional learning methods (Hu et al., 2022; Han et al., 2024) often struggle to preserve existing capabilities during such situations. Continual learning enables LLMs to flexibly integrate new and existing knowledge as tasks increase, addressing the limitations of static training in preserving prior performance while incorporating new information. The core challenge lies in achieving an optimal balance between plasticity and stability (Mermillod et al., 2013). Excessive plasticity will result in catastrophic forgetting, whereas overly strong stability may prevent efficient and effective knowledge transfer.

---

[*]Equal contribution
[†]Corresponding authors.

39th Conference on Neural Information Processing Systems (NeurIPS 2025).

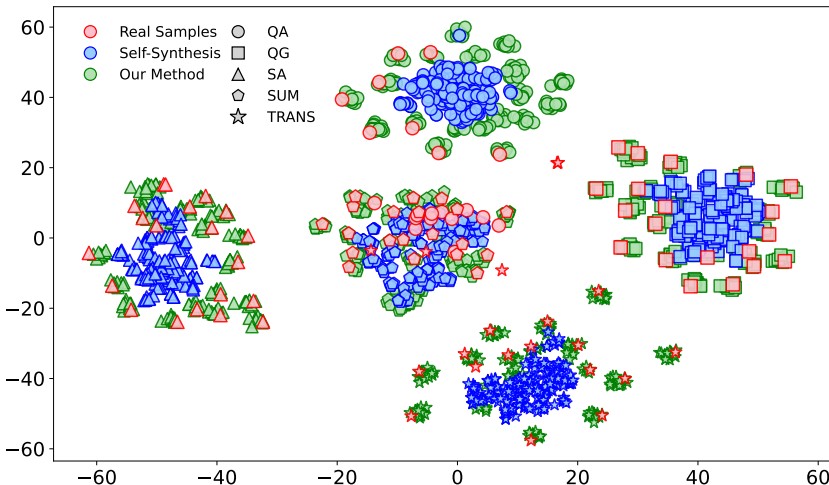

Figure 1: Clustering analysis of pseudo samples generated by our method (SERS) and Self-Synthesized Rehearsal (SSR) approaches across five tasks, alongside real samples. It can be observed that the pseudo samples generated by our method are closer to the real samples than SSR, indicating that SERS produces **more similar** pseudo samples that better reflect knowledge from previous tasks.

A series of works (Zhao et al., 2024; Zheng et al., 2025a; Wang et al., 2024a; Sun and Gao, 2024) have been proposed to mitigate this balancing challenge. Rehearsal-based methods (Yin et al., 2022; de Masson D'Autume et al., 2019; Rolnick et al., 2019) preserve model capabilities on previous tasks by utilizing real samples from prior training processes, which are not always consistently available in practice. To tackle the challenge of limited access to get historical data, existing solutions (Sun et al., 2020; Zhao et al., 2024) apply pseudo-sample generation, yet the additional generation modules increase the number of trainable parameters. Huang et al. (2024) leverage in-context learning capacity of LLMs for self-synthesis rehearsal, effectively alleviating parameter burdens. We unexpectedly found that self-synthesized samples often exhibit low similarity to real data, failing to adequately reflect the knowledge structure and thus undermining the effectiveness of rehearsals, as shown by the clustering analysis in Figure 1. Regularization-based approaches (Guo et al., 2024; Wang et al., 2023) impose constraints on loss functions to penalize parameter updates that affect prior task knowledge. However, traditional static constraint methods, with their fixed trade-off between facilitating knowledge transfer and preventing forgetting, lack consideration for task diversity.

To generate pseudo samples that better support knowledge consolidation during rehearsal, while flexibly balancing knowledge transfer and forgetting prevention across tasks, we propose **S**elf-**E**volving Pseudo-**R**ehearsal with Task **S**imilarity (SERS). Specifically, we generate pseudo inputs using template guidance and semantic masking, eliminating task-specific instructions, where dynamic guidance and mask ratios ensure the diversity. After generating the pseudo inputs, to supplement pseudo labels, over-specialized samples are selected via label self-questioning and ease the demand for task-specific knowledge through label self-evolution. In the rehearsal stage, to fully promote permissible knowledge transfer, we design a regularization loss function based on task similarity. When tasks are similar, the regularization is relaxed to encourage the integration of new and old knowledge; otherwise, constraints are strengthened to alleviate the forgetting of previous knowledge.

We conducted extensive experiments on the SuperNI dataset (Wang et al., 2022) using LLaMA2-7B (Touvron et al., 2023) and ChatGLM-6B (GLM et al., 2024) to evaluate the performance of SERS across varying task chains. Results show that SERS consistently outperforms existing methods and is more stable across a variety of task orders. On LLaMA2-7B, it achieved a 2.16% relative improvement over advanced pseudo-sample rehearsal approaches, closely matching Multi-Task Learning (MTL) performance; on ChatGLM-6B, it even surpassed MTL.

The main contributions of our work are as follows:

- We propose SERS, a continual learning framework for LLMs that decouples input and label synthesis. SERS generates pseudo inputs via template guidance and semantic masking and uses a

label self-evolution module to prevent over-specialization in pseudo labels, brings human learning strategies into machine learning.

- We introduce a task similarity-based dynamic regularization to effectively balance stability and plasticity, reducing the sensitivity of knowledge transfer to task order.

- Experiments show that SERS significantly improves learning accuracy under task-incremental conditions, alleviates catastrophic forgetting, and even facilitates additional knowledge transfer.

## 2 Related Work

### 2.1 Self-Evolution Learning

Self-evolution (Zhong et al., 2022; Peng et al., 2023; Zhong et al., 2023; Zheng et al., 2023; Tao et al., 2024; Song et al., 2025) is a paradigm that enables models to learn and improve through self-generated knowledge, inspired by human learning from experience. In this process, LLMs create new tasks and solutions based on predefined goals, collect feedback from the environment, refine the acquired experience to eliminate errors, and update their parameters or context accordingly.

Zhong et al. (2022) improve pretraining efficiency through a two-stage process of self-questioning and self-evolution. In the first stage, the model uses masking to detect tokens it struggles to understand; in the second, it generates soft labels with richer knowledge patterns to enhance training. Similarly, Singh et al. (2023) apply reinforcement learning to actively generate new samples, evaluate them using a binary reward function, and select high-quality data for model updates.

Motivated by these strategies and aiming to address the instability of pseudo-sample generation in LLMs, we propose a label-level self-evolution method. By imitating self-questioning and self-evolution structure, our approach detects over-specialized pseudo labels and smooths them with general knowledge, mitigating the local overfitting to a specific task caused by rehearsal.

### 2.2 Continual Instruction Tuning for LLMs

Continual instruction tuning for LLMs extends traditional LLMs tuning by enabling LLMs to incrementally absorb new tasks and feedback without forgetting prior knowledge. Compared to standard continual learning, it introduces unique challenges due to generative outputs, global semantic relationships, and model scale. Existing methods can be broadly categorized into:

(1) Architecture-based (Ren et al., 2024; Zhao et al., 2024; Ke et al., 2023): These methods adjust model architecture or parameter distribution to separate the knowledge of new and old tasks, thus mitigating forgetting caused by parameter interference. For example, Ren et al. (2024) use fast and slow learners to balance stability and plasticity. Zhao et al. (2024) introduce an Attentive Learning & Selection module by combining multiple PET blocks in different ways to fit different tasks. However, as task numbers increase, adding new modules raises computational costs, and separate architecture adjustments limit flexibility and universality.

(2) Rehearsal-based methods (Wang et al., 2024b; Huang et al., 2024; Maekawa et al., 2023): These methods involve real or pseudo-sample rehearsal. Real sample rehearsal, as in Wang et al. (2024b), helps recall past knowledge but relies on access to original data during each training stage, which is often impractical. Pseudo-sample rehearsal typically necessitates an extra generation module, increasing trainable parameters. To our knowledge, Huang et al. (2024) are the first to use self-synthesis to generate pseudo samples from a few real samples, solving storage issues, but still facing challenges with pseudo samples instability and task comprehension.

(3) Regularization-based methods (Wang et al., 2023; Jin et al., 2021; Li et al., 2024; Guo et al., 2024): These methods constrain excessive parameter updates with regularization. For example, Wang et al. (2023) learn tasks in different low-rank vector subspaces and keep these subspaces orthogonal to minimize interference. However, orthogonal subspaces limit the knowledge transfer between tasks.

Our method combines pseudo-sample rehearsal and regularization. For pseudo-sample generation, we leverage template guidance and semantic masking to ensure the stability and real-sample similarity of synthesized pseudo samples, while varying templates and masking ratios promote diversity. For regularization, we dynamically adjust the regularization strength based on task similarity and account

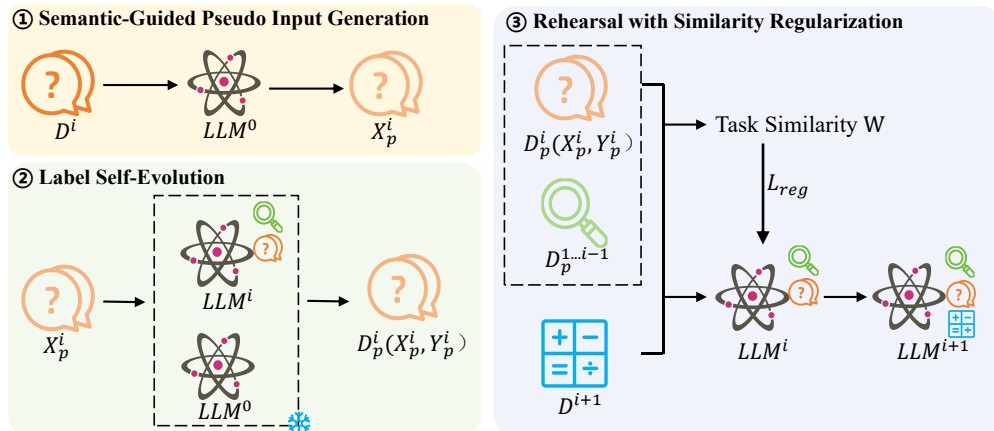

Figure 2: The overall framework of our SERS method. In the Semantic-Guided Pseudo-Input Generation stage, a small set of real samples produces pseudo inputs $X_p^i$. Then, the Label Self-Evolution module refines these inputs by integrating knowledge from $LLM^0$ and $LLM^i$, yielding rehearsal pseudo samples $D_p^i(X_p^i, Y_p^i)$. Finally, in the Rehearsal with Similarity Regularization stage, the rehearsal samples and the new training data $D^{i+1}$ are combined for fine-tuning, with regularization applied based on task similarity.

for the impact of task order on parameter updates, effectively balancing knowledge transfer and resistance to catastrophic forgetting.

## 3 Methodology

### 3.1 Problem Definition

We consider the problem of Task-Incremental Continual Instruction Tuning. Given a sequence of $N$ instruction-following tasks $\mathcal{T}_1, \mathcal{T}_2, \ldots, \mathcal{T}_N$, each associated with a dataset $\mathcal{D}i = (X^i, Y^i)$, the goal is to continually fine-tune a pre-trained language model $LLM_0$ on these tasks in sequence. At each step $i$, the model receives only the current task dataset $\mathcal{D}_i$ and fine-tunes the model $LLM^{i-1}$ to obtain $LLM^i$. The objective is to learn each new task while maintaining performance on all previous tasks, without requiring large-scale retraining.

### 3.2 Framework Overview

In this paper, we propose a continual learning framework for LLMs that combines pseudo-sample rehearsal with regularization. As shown in Figure 2, our approach consists of three main components: semantic-guided pseudo-input generation, label self-evolution, and rehearsal with similarity regularization. In the following sections, we provide a detailed explanation of each module.

### 3.3 Semantic-Guided Pseudo-Input Generation

The self-synthesis approach proposed by Huang et al. (2024) effectively addresses the limitations discussed above, but still faces key challenges: the generated pseudo samples cannot well reflect the original knowledge structure and thus provide limited support for rehearsal. Additionally, appending task instructions and labels increases the model's comprehension burden, while the generated labels lose meaning after further refinement. To address these problems, we propose a Semantic-Guided Pseudo-Input Generation module. As shown in Figure 3, real examples are masked in two roles: as Example Template that providing structure guidance, and as Semantic Guidance that offering semantic context. Experiments in Wang et al. (2024c) indicate that models not fine-tuned on a specific task have stronger contextual understanding, so that we use $LLM^0$ to fill in the masks to get pseudo inputs $X_p^i$ without extra generating block. Varying mask ratios and example templates enhance diversity, while removing task instructions and labels reduces cognitive load. Figure 1 shows the clustering of pseudo samples from our method, the self-synthesis approach, and real data. Our pseudo

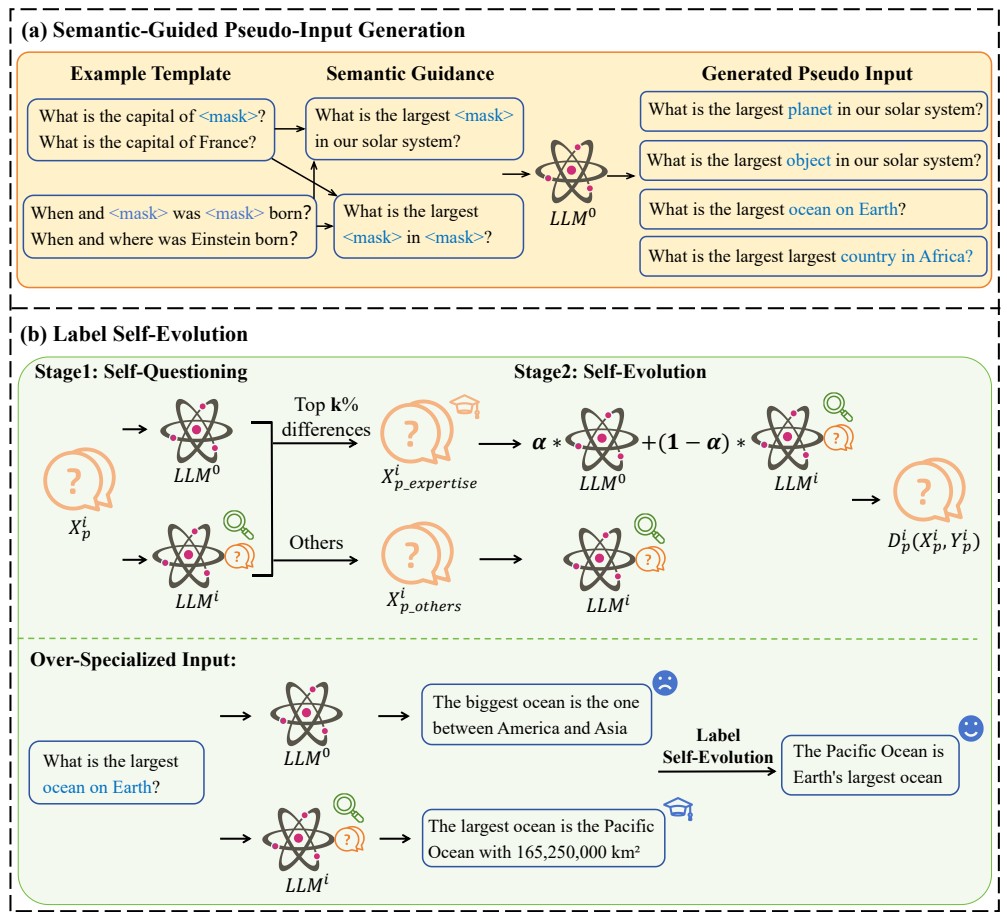

Figure 3: Detailed illustration of the core modules. **(a) Semantic-Guided Pseudo-Input Generation:** A real sample and its masked version serve as an example template, while an additional masked sample provides semantic guidance. These are combined and passed through $LLM^0$ to generate pseudo inputs. **Varying combinations and mask ratios** promote diversity. **(b) Label Self-Evolution:** (Top) In self-questioning stage, the top-$k\%$ pseudo inputs with high domain dependence are identified as **over-specialized samples**. In self-evolution stage, these are relabeled by blending knowledge from $LLM^0$ and $LLM^i$; others directly use the output of $LLM^i$ as labels. (Bottom) An example shows that an over-specialized input leads to **poor** output from $LLM^0$ and an **overly detailed** output from $LLM^i$. After label self-evolution, the final label becomes more **acceptable**, with reduced reliance on domain-specific knowledge.

samples are more similar to real ones, preserving knowledge structure while maintaining diversity. Examples in real-task settings are shown in Appendix A.

## 3.4 Label Self-Evolution

Considering that randomness in pseudo-input generation can lead to over-specialized instances requiring excessive expertise, rehearsing with such samples may cause large parameter shifts and disrupt existing knowledge. We therefore introduce a label self-evolution method inspired by human review. As shown at the top of Figure 3, the process consists of two stages: self-questioning and self-evolution. In self-questioning, both the base model $LLM^0$ and fine-tuned model $LLM^i$ generate labels. Samples with the top-$k\%$ output differences are treated as over-specialized. In the self-evolution stage, regular samples adopt $LLM^i$'s outputs as labels, while over-specialized samples are relabeled by integrating the outputs from both models using a weighted combination. The coefficient $\alpha$ adjusts the contributions of $LLM^0$ and $LLM^i$ to balance general and task-specific knowledge.

As shown in the lower part of Figure 3, over-specialized inputs produce vague outputs on $LLM^0$ and highly specific ones on $LLM^i$. The label self-evolution module merges these to produce acceptable labels, mitigating overfitting during rehearsal. Examples in Figure 8 illustrate the effectiveness and reliability of this process on real tasks.

## 3.5 Rehearsal with Similarity Regularization

This section explores how task similarity, reflected through task order, influences training results. After generating pseudo samples avoiding over-specialization, pseudo samples are used for rehearsal training. At the training stage of $T^{i+1}$, the model is updated using pseudo samples of old tasks $D_p^{1\dots i-1}$, $D_p^i$ and new task training data $D^{i+1}$. Previous works (Huang et al., 2024; Zhao et al., 2024) fine-tune using LoRA (Hu et al., 2022) without adapting to task characteristics, making the results highly sensitive to task order. Since the model already contains knowledge from earlier tasks, similar new tasks should allow more knowledge transfer, whereas dissimilar ones require stronger constraints to maintain prior knowledge. To achieve this, we design a regularization loss based on task similarity, which is incorporated into the original cross-entropy loss to adjust LoRA fine-tuning:

$$L = L_{ce} + \lambda \cdot L_{\text{reg}}, \tag{1}$$

where $L_{ce}$ is the standard cross-entropy loss, $\lambda$ controls regularization strength, and $L_{reg}$ is the regularization term, balancing knowledge sharing and parameter stability. Specifically, we impose the regularization constraint on all training target parameters during optimization, as formulated in Equation 2:

$$L_{\text{reg}} = \frac{1}{2} \sum_i \mathbb{E}\big[\|\theta_i\|^2\big]. \tag{2}$$

We modulate regularization strength $\lambda$ based on task similarity $W$ and rehearsal ratio $r_{replay}$. When $r_{replay}$ is low or $W$ is large, indicating limited rehearsal or low task similarity, stronger constraints are applied to stabilize knowledge retention. In contrast, higher $r_{replay}$ or smaller $W$ suggests task alignment, allowing more relaxed regularization to facilitate knowledge transfer. This dynamic adjustment enables SERS to maintain a balance between stability and plasticity, supporting more robust continual learning. Accordingly, $\lambda$ is formally defined in Equation 3:

$$\lambda = \Big[\lambda_{\min} + \big(\lambda_{\max} - \lambda_{\min}\big)\Big(1 - e^{-\frac{W}{W_{\text{th}}}}\Big)\Big] * \big(1 - r_{\text{replay}}\big), \tag{3}$$

where $\lambda_{\min}$ and $\lambda_{\max}$ control the range of regularization strength, and $W_{\text{th}}$ adjusts the curvature of the scaling function. The Wasserstein Distance (Chen et al., 2022; Liu et al., 2025), a representative of the optimal transport framework (Alvarez-Melis and Fusi, 2020), provides a metric for assessing the similarity between the distributions of two datasets, which is defined as in Equation 4:

$$W(P, Q) = \inf_{\gamma \in \Pi(P,Q)} \mathbb{E}_{(X,Y)\sim\gamma} \left[\|X - Y\|\right]. \tag{4}$$

# 4 Experiment

## 4.1 Dataset and Metrics

Our experiments are conducted on the SuperNI (Wang et al., 2022) dataset, a large and comprehensive benchmark for instruction tuning. For fair comparison, we adopt the same ten tasks as Huang et al. (2024), divided into two groups: one with five tasks and the other with ten. Each group is evaluated under three different task orders. Experiments were carried out on LLaMA2-7B(Touvron et al., 2023) and ChatGLM-6B(GLM et al., 2024). For more details about the dataset, please refer to Appendix C.

We adopt the Rouge-L score (Lin, 2004) to assess generation quality, where $R_j^i$ denotes the model's performance on task $j$ at stage $i$. To evaluate overall performance, knowledge transfer ability, and retention of prior knowledge, the following commonly used continual learning metrics are selected:

**(1) Average Rouge-L (AR).** After training on the final task, the average performance across all tasks is computed as shown in Equation 5:

$$AR = \frac{1}{N} \sum_{i=1}^{N} R_i^N. \tag{5}$$

Table 1: Results on LLaMA2-7B and ChatGLM-6B under different task orders and settings.

| Model | Order 1 | | Order 2 | | Order 3 | | Avg. | |
|---|---|---|---|---|---|---|---|---|
| | $AR \uparrow$ | $BWT \uparrow$ | $AR \uparrow$ | $BWT \uparrow$ | $AR \uparrow$ | $BWT \uparrow$ | $AR \uparrow$ | $BWT \uparrow$ |
| **LLaMA2-7B 5Tasks** | | | | | | | | |
| MTL | 53.07 | – | 53.07 | – | 53.07 | – | 53.07 | – |
| KMeansSel(1%) | 49.73 | -5.22 | 50.14 | -4.17 | 50.12 | -3.61 | 50.00 | -4.33 |
| L2 | 28.62 | -28.99 | 29.22 | -28.45 | 28.33 | -30.71 | 28.72 | -29.38 |
| SAPT | 50.47 | -3.75 | 51.04 | -2.98 | 50.22 | -4.37 | 50.58 | -3.70 |
| SSR | 51.33 | -1.97 | 52.41 | -1.18 | 52.02 | -1.01 | 51.92 | -1.39 |
| **SERS** | **52.90** | **-0.55** | **53.01** | **-0.27** | **52.84** | **-0.63** | **52.92** | **-0.48** |
| **ChatGLM-6B 5Tasks** | | | | | | | | |
| MTL | 48.92 | – | 48.92 | – | 48.92 | – | 48.92 | – |
| KMeansSel(1%) | 43.72 | -5.64 | 43.74 | -5.07 | 45.13 | -4.37 | 44.19 | -5.02 |
| L2 | 25.19 | -35.32 | 26.46 | -32.47 | 26.18 | -34.92 | 25.94 | -34.24 |
| SAPT | 49.01 | -1.54 | 48.65 | -2.11 | 49.23 | -1.89 | 48.96 | -1.84 |
| SSR | 48.95 | -2.12 | 49.02 | -1.94 | 49.38 | -0.51 | 49.11 | -1.52 |
| **SERS** | **49.97** | **-0.89** | **49.86** | **-1.17** | **50.04** | **-0.48** | **49.98** | **-0.85** |
| **LLaMA2-7B 10Tasks** | | | | | | | | |
| MTL | 64.72 | – | 64.72 | – | 64.72 | – | 64.72 | – |
| KMeansSel(1%) | 59.13 | -5.88 | 60.71 | -5.39 | 60.44 | -7.17 | 60.09 | -6.15 |
| L2 | 33.13 | -28.99 | 34.71 | -25.12 | 37.02 | -22.71 | 34.95 | -25.42 |
| SAPT | 62.51 | -2.06 | 61.90 | -2.81 | 62.29 | -2.30 | 62.23 | -2.39 |
| SSR | 62.29 | -1.84 | 62.64 | -1.86 | 62.36 | -3.95 | 62.43 | -2.55 |
| **SERS** | **63.42** | **-1.72** | **63.45** | **-1.11** | **64.46** | **-2.27** | **63.78** | **-1.7** |
| **ChatGLM-6B 10Tasks** | | | | | | | | |
| MTL | 62.04 | – | 62.04 | – | 62.04 | – | 62.04 | – |
| KMeansSel(1%) | 60.84 | -5.41 | 61.24 | -4.77 | 61.04 | -5.27 | 61.04 | -5.15 |
| L2 | 40.18 | -29.71 | 41.37 | -28.66 | 41.99 | -26.12 | 41.18 | -28.16 |
| SAPT | 61.30 | -3.44 | 61.56 | -2.73 | 60.87 | -2.92 | 61.24 | -3.03 |
| SSR | 62.68 | -1.79 | 62.27 | -2.42 | 61.80 | -1.56 | 62.25 | -1.92 |
| **SERS** | **63.30** | **-1.17** | **63.22** | **-1.52** | **63.16** | **-1.49** | **63.23** | **-1.39** |

**(2) Backward Transfer (BWT).** BWT measures the degree to which the learning of subsequent tasks affects the performance of the learned tasks, which is defined as:

$$BWT = \frac{1}{N-1} \sum_{i=1}^{N-1} (R_i^N - R_i^i). \tag{6}$$

## 4.2 Experiment Details

All experiments were conducted on a single A100 GPU. For pseudo-sample generation, 1% of real samples are used to create pseudo samples equivalent to 10% of the training data. In the self-questioning stage, we set $k = 20$, $\alpha = 0.5$ for LLaMA2-7B, and $k = 10$, $\alpha = 0.6$ for ChatGLM-6B to control the selection and evolution of over-specialized samples. LoRA is used for fine-tuning, and Wasserstein distance based on model embeddings guides the regularization process.

## 4.3 Experiment Results

We compare our SERS method with several representative baselines, including the classic rehearsal-based KMeansSel, which selects real samples via KMeans clustering; the advanced pesudo-sample rehearsal approach SSR (Huang et al., 2024), which leverages self-synthesis to generate pseudo samples for rehearsal; advanced structure-based method SAPT (Zhao et al., 2024), which employs a Shared Attentive Learning & Selection module to align the PET learning and selection; and the regularization-based L2 method. A multi-task learning (MTL) baseline, which jointly trains all tasks without considering forgetting, is also included for reference. As shown in Table 1, SERS consistently

Table 2: The ablation studies on each proposed module. SGG refers to our Semantic-Guided Pseudo-Input Generation. LSE denotes Label Self-Evolution strategy. SR represents Similarity Regularization. A "✓" indicates that our module is applied, while a"–" denotes the use of a corresponding strategy from existing advanced pseudo-rehearsal approaches.

| Ablation Setting | | | LLaMA-7B AR (%) ↑ | | | ChatGLM-6B AR (%) ↑ | | |
|---|---|---|---|---|---|---|---|---|
| SGG | LSE | SR | Order 1 | Order 2 | Order 3 | Order 1 | Orde r2 | Order 3 |
| – | – | – | 51.33 | 52.41 | 52.02 | 48.95 | 49.02 | 49.38 |
| ✓ | – | – | 52.37 | 52.63 | 52.54 | 49.87 | 49.58 | 49.35 |
| – | ✓ | – | 51.99 | 52.83 | 52.33 | 49.25 | 49.40 | 49.02 |
| – | – | ✓ | 52.40 | 52.42 | 52.61 | 49.68 | 49.62 | 49.50 |
| ✓ | ✓ | – | 52.47 | 52.78 | 52.71 | 49.78 | 49.33 | 49.80 |
| – | ✓ | ✓ | 52.56 | 52.92 | 52.73 | 49.08 | 49.43 | 49.21 |
| ✓ | – | ✓ | 52.59 | 52.95 | 52.56 | 49.70 | 49.80 | 49.62 |
| ✓ | ✓ | ✓ | **52.90** | **53.01** | **52.84** | **49.97** | **49.86** | **50.04** |

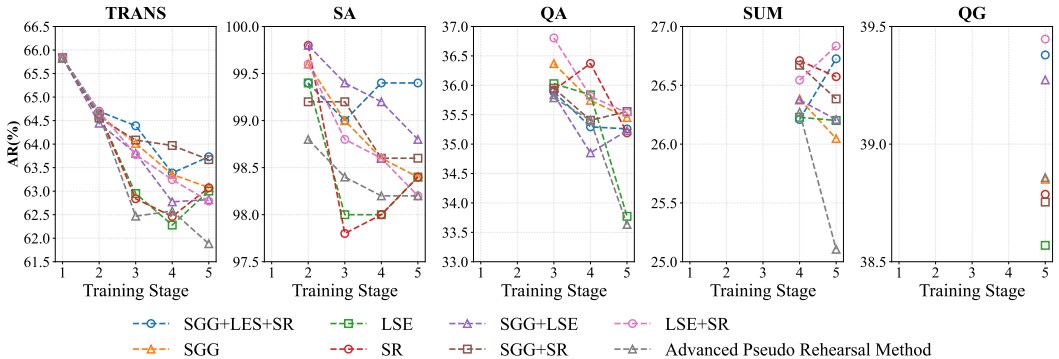

Figure 4: Ablation results detailing the performance variations of the LLaMA2-7B model across a 5tasks sequence under Order 1 (TRANS → SA → QA → SUM → QG). More details of ablation results are shown in Figure 9

outperforms these baselines and maintains stable performance across task orders, demonstrating the effectiveness of similarity regularization. On LLaMA2-7B, it achieves results close to MTL, surpassing the next best method by 2.16% and exhibiting significantly lower BWT. Notably, in ChatGLM-6B, where MTL suffers from task confusion due to global attention and 2D positional encoding, SERS surpasses MTL by incrementally refining decision boundaries through rehearsal with staged updates.

# 5  Ablation and Comparison Experiments

## 5.1  Module Ablation

In this section, we carry out ablation studies to verify the effectiveness of each module. All experiments are on 5tasks, and we measure performance with the AR metric. The results appear in Table 2, and detailed ablation results are provided in Figure 4. SERS introduces three core improvements: Semantic-Guided Generation, Label Self-Evolution, and Similarity Regularization. We evaluate the effectiveness of these components with different configuration settings. For settings that do not include the SERS modules, we adopt corresponding strategies from Huang et al. (2024), where pseudo samples are generated via in-context learning, pseudo labels are directly refined on the task-specific model, and no regularization method is applied. The settings with semantic-guided generation achieve higher overall performance compared to those using existing advanced method to generate pseudo samples. The curves with task similarity are more likely to exhibit task-level performance

improvement during training, and those incorporating label self-evolution tend to perform better on new tasks, demonstrating the effectiveness of our proposed improvements.

## 5.2 Data Utilization Efficiency

In our experiments, we first employed 1% of real samples to synthesize 10% pseudo samples, achieving strong performance with efficient data utilization. We then extended the analysis by generating different amounts of pseudo samples for rehearsal using various proportions of real data (1%, 0.75%, 0.5%, and 5%) to investigate the trade-off between data efficiency and pseudo-sample redundancy. As shown in Figure 5, even a small number of real samples can produce pseudo samples that are diverse and capable of capturing the underlying task knowledge. However, as the pseudo-sample ratio increases, the improvement in AR gradually saturates. When generating 20% pseudo samples from 1% real data, performance begins to decline due to excessive redundancy and interference with learning new tasks.

When further reducing the number of real samples, we observe that using 0.75% or 0.5% of real data yields slightly better performance than 1% real data when synthesizing a small proportion of pseudo samples. Nevertheless, the performance degrades notably as the pseudo-sample ratio grows. This suggests that with a small pseudo-sample ratio, fewer real samples can better capture the essential task knowledge and improve synthesis quality. In contrast, when a larger number of pseudo samples are generated, the limited diversity of real samples leads to higher redundancy, which hinders learning effectiveness. Moreover, pseudo samples generated from a larger real dataset tend to form more cluster centers. Under low pseudo-sample ratios, this results in less coherent knowledge structures and slightly worse performance than using 1% real data. Yet, as the pseudo-sample ratio increases to 20%, the performance improves substantially and approaches that of multi-task learning (MTL).

We also compare pseudo-sample rehearsal with real-sample rehearsal, as presented in Table 5 of the Appendix E. The results are consistent with findings from SSR, showing that even when 10% of real samples are replayed, the performance remains inferior to that of pseudo-sample rehearsal. This is because labels synthesized by the old model facilitate learning, improving the new model's task adaptation. In contrast, real-sample rehearsal is directly constrained by the limited proportion of available real data, whereas pseudo-sample rehearsal can flexibly expand data diversity by synthesizing new samples from a fixed real set. Consequently, the 1% real-sample rehearsal fails to match the performance achieved with 5% real samples, as the smaller rehearsal ratio restricts the model's ability to preserve prior knowledge.

## 5.3 Analysis of Parameters

We analyze the impact of two key parameters in label self-evolution. The proportion threshold $k$ determines the selection of over-specialized samples during self-questioning. As shown in Figure 6, setting $k$ too high omits valuable specialized knowledge, while a low $k$ allows too many over-specialized samples for rehearsal, causing bias in model parameters and affecting overall performance. $\alpha$ controls the balance between general ($LLM^0$) and task-specific ($LLM^i$) knowledge when refining labels. Figure 6 illustrates that a high $\alpha$ may tend to less accurate labels, whereas a low $\alpha$ reduces the smoothing effect, weakening the integration of general and specialized knowledge.

Parameter adjustment should consider model capability. Stronger mask-filling models better preserve prior knowledge in pseudo samples, allowing for a smaller $k$; weaker models require a larger $k$ to avoid excessive specialization. Similarly, models with stronger downstream abilities benefit from a higher $\alpha$ to increase the general knowledge in over-specialized samples, while less capable models require a lower $\alpha$ to avoid inaccurate labels.

## 6 Conclusion

In this work, we propose **SERS** for catastrophic forgetting mitigation in LLMs. SERS generates pseudo samples that better reflect the structural knowledge of previous tasks, prevents over-specialization on rehearsal pseudo samples from harming overall performance and dynamically adjusts regularization strength based on the similarity between previous and new tasks. Extensive experiments demonstrate that, compared to various representative methods, SERS achieves more

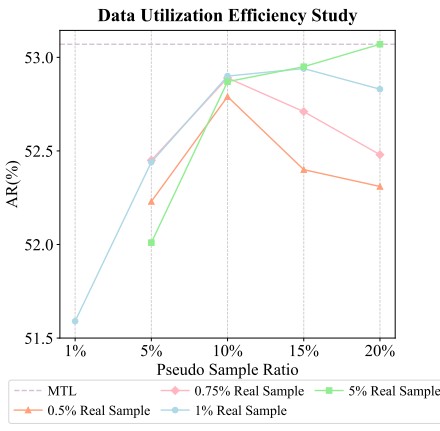

Figure 5: **Rehearsal Analysis.** We generate various proportions of pseudo samples using different amounts of real samples ranging from 0.5% to 5% on LLaMA2-7B to evaluate data utilization efficiency.

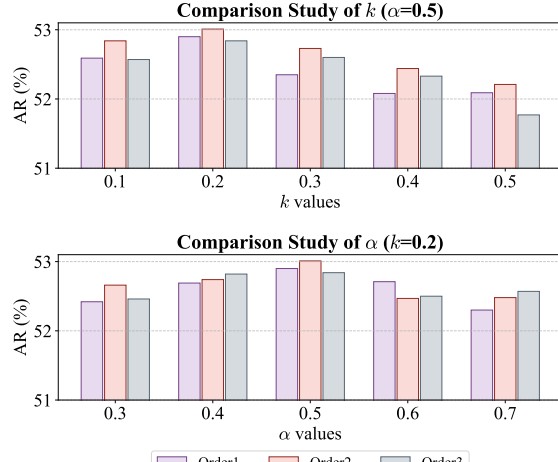

Figure 6: **Parameter Analysis.** We evaluate SERS performance on LLaMA2-7B by varying $k$ and $\alpha$ while fixing the other parameter respectively.

effective forgetting mitigation and enhanced performance stability, underscoring SERS's potential as a general solution for continual learning in LLMs.

## Limitations

Although ablations confirm each module's contribution to AR and show score performance during training, the complex relationships between tasks make it hard to pinpoint why some tasks improve or decline. A deeper analysis of how new tasks affect previous tasks may unlock further gains in continual accuracy. Moreover, while pseudo-sample rehearsal boosts review of past knowledge, it remains unclear whether these synthetic examples can introduce knowledge beyond the original data. Exploring the ability of pseudo samples to enrich the model with unseen knowledge could be key to surpassing MTL in future continual learning work.

## Acknowledgments and Disclosure of Funding

This work is supported by the National Key Research and Development Program of China (2023YFC2705700), the National Natural Science Foundation of China (Grant No. 62225113, U23A20318, U2336211 and 62276195), the Foundation for Innovative Research Groups of Hubei Province (Grant No. 2024AFA017) and the Science and Technology Major Project of Hubei Province (Grant No. 2024BAB046). The numerical calculations in this paper have been done on the supercomputing system in the Supercomputing Center of Wuhan University.

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

# A   Comparative Case Study on Pseudo-Sample Generation

To illustrate the effectiveness of our proposed method, we present a comparison between the traditional ICL approach and the Semantic-Guided Pseudo-Input Generation we introduced. Using two samples from the same task, we generate pseudo data in two ways for two retries: the ICL method generates both inputs and outputs with 2-shot prompting, while our method focuses on generating only pseudo inputs. As shown in Figure 7, it is evident that the 2-shot ICL approach fails to capture the essence of the "cosmosqa" task. The generated questions resemble generic QA samples, lacking the implicit cues that emphasize common-sense reasoning. In contrast, our method is able to produce samples that closely resemble real ones under a low masking ratio. When the masking ratio is high, the generated outputs are diverse but still retain the core features of the "cosmosqa" task. This not only ensures the diversity of pseudo samples, but also supports the model's review of previous knowledge.

---

**Real Sample:**

**Input:**
Emmy isn't feeling well. I woke up this morning with a little bit of vom on the bed near her mouth. Dog's puke all the time though, so no big deal. I let her out and she took a big old solid 2-point conversion. But she didn't run or scuff her feet afterwards, she just ambled back inside.
Question: What did I mean when I said she took a solid 2-point conversion?
**Output:** She pooped.

**Input:**
blaine was only able to stay for a little while since he had to work sunday and while i wanted to go home with him to NOT face the party the rest of the evening, i stayed with nick and brenna. i had such a good time though. there was making out involved among some of the party guests. oh, might i add that the party took place on the rooftop of this lovely apartment building.
Question: Where might James have gone after leaving the party?
**Output:** He went home.

---

**In-Context Learning Output:**

**Input:**
The best thing about being single is you can eat whatever you want whenever you feel like eating it!
Question: When would Katie most likely be thinking these thoughts?
**Output:** After having an argument over dinner at someone else's house.

**Input:**
Whenever we meet up at a cafe or something, he always seems very keen to see me again, and wants us to do things together all the time! But then when I'm ready to leave, his eyes get all glassy and sad looking and he says Ï don't want you to think badly about me.Ït makes me feel like maybe my presence isn't actually welcome - I can tell he doesn't really enjoy being around me, even though he tries hard to hide it.
Question: What does the speaker most likely think about herself?
**Output:** She thinks highly of herself.

**Semantic-Guided Generation:**

**Input:**
Blaine was reluctant to leave early because he had to work Sunday and wanted to spend more time with Nick and Brenna. He decided not to go home with them but instead to return later tonight. I was having such a good time at the party as were many others who made it into the exclusive venue located on top of this amazing apartment complex. Oh, might I add that the party location was right outside our living room window!
Question: Where else could James have gone after leaving the party?

**Input:**
Blaine seemed like such a nice guy since it came down to him being the last person standing at the bar Sunday night. He wanted me to stay for just a little longer; however, I didn't want to face the remainder of the evening alone. So I went home with Nick and Brenna instead. They were very kind and offered us plenty of space even if we made our own drinks or snacks from what they provided. Oh! And by the way, their roof-top patio was absolutely gorgeous!
Question: What would you think about James based upon his behavior

Figure 7: A case study for Pseudo-Sample Generation

# B   Comparative Case Study on Output Refinement

This part demonstrates the value of self-evolution in label refinement through real examples. By adjusting the balance between general knowledge from $LLM^0$ and task-specific knowledge from $LLM^i$, the model can reduce overly specialization and produce more acceptable responses. In the QA example shown in Figure 8, the output from $LLM^0$ contains hallucinations and fails to provide a correct answer. In contrast, the output from $LLM^i$ correctly identifies the key term "family". The final response refined through self-evolution also delivers a precise and reasonable answer.

---

**Input:** A small part of her face lit up when she heard the news. Her eyes glistened but there wasn't any trace of joy on them or happiness even though this was great news. She simply looked relieved because finally things were starting to go well for her family. And they had been through hell lately so every little thing counted nowadays. So she couldn't help feeling grateful.
**Question:** What made the person feel relief?

$LLM^0$ **Output:** Things went wrong recently!
$LLM^i$ **Output:** They got good news for their family.
**Output after Self-Evolution:** The person was told that things was going to be okay.

---

Figure 8: A case study for output refinement

# C   Additional Experimental Details

All samples are selected from the SuperNI dataset. To ensure fair comparison, we follow the same task settings as advanced pseudo-sample rehearsal methods (Huang et al., 2024). The selected tasks are shown in Table 3. During pseudo-sample generation, we begin with 1% of the real samples and apply the Semantic-Guided Pseudo-Input Generation method with three retries to create a large number of pseudo samples. Then, we use the K-Means algorithm to select 10% of these for rehearsal. During the training phase, we selected two numbers of tasks and two task sequences to form four task chains, with the task chains as shown in Table 4.

Table 3: Details of task names and abbreviations

| Abbreviation | Task Name |
|---|---|
| QA | task024_cosmosqa_answer_generation |
| QG | task074_squad1.1_question_generation |
| SA | task1312_amazonreview_polarity_classification |
| SUM | task511_reddit_tifu_long_text_summarization |
| TRANS | task1219_ted_translation_en_es |
| DSG | task574_air_dialogue_sentence_generation |
| EXPL | task192_hotpotqa_sentence_generation |
| PARA | task177_para-nmt_paraphrasing |
| POS | task346_hybridqa_classification |
| PE | task064_all_elements_except_first_i |

Table 4: Details of Task Chains under Different Task Numbers and Orders

| Settings | Task Chain |
|---|---|
| 5Tasks Order 1 | TRANS → SA → QA → SUM → QG |
| 5Tasks Order 2 | QA → QG → SA → SUM → TRANS |
| 5Tasks Order 3 | SUM → QG → TRANS → QA → SA |
| 10Tasks Order 1 | TRANS → SA → QA → SUM → QG → PE → PARA → POS → DSG → EXPL |
| 10Tasks Order 2 | QA → QG → SA → SUM → TRANS → DSG → EXPL → PARA → PE → POS |
| 10Tasks Order 3 | SUM → QG → TRANS → QA → SA → PARA → DSG → POS → EXPL → PE |

# D  Ablation Study Details

In this section, we present the detailed performance of each ablation setting across different task chains and models. The results show that our generation strategy consistently leads to better outcomes, label self-evolution generally benefits the learning of new tasks, and similarity regularization facilitates knowledge transfer, increasing the likelihood of performance gains throughout training. Details are shown in Figure 9.

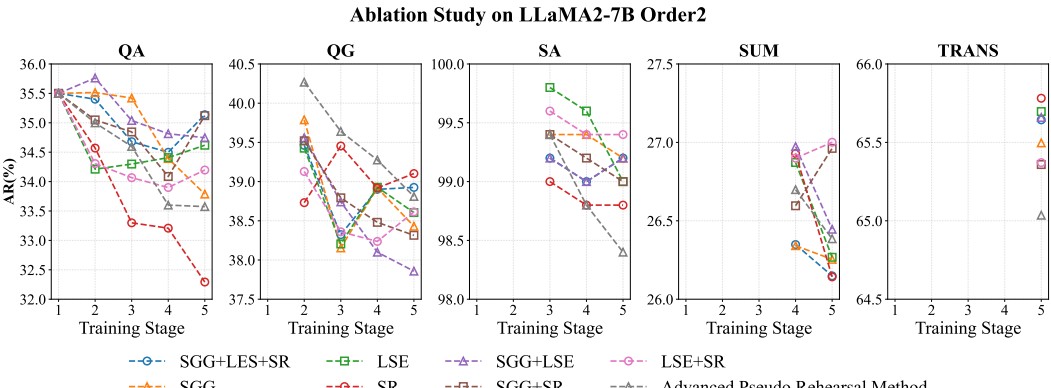

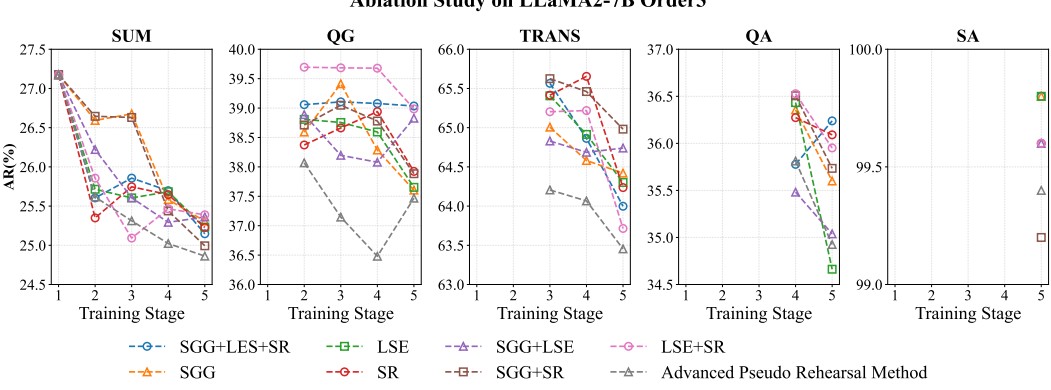

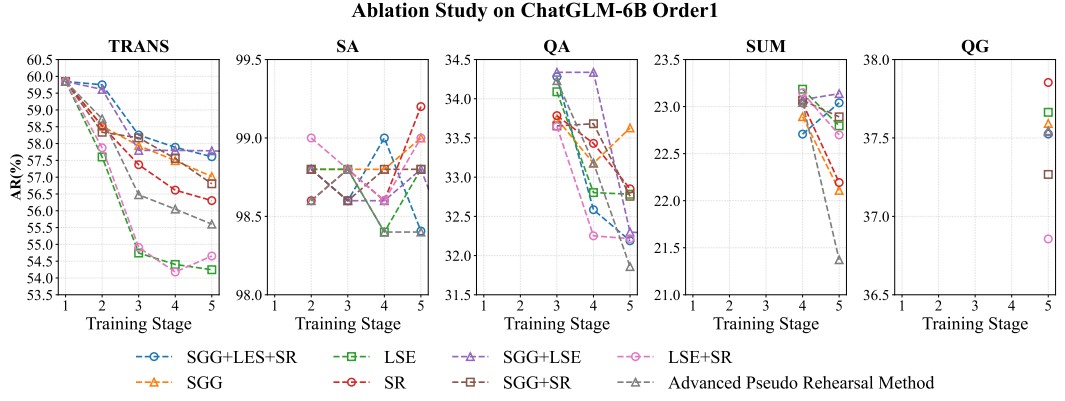

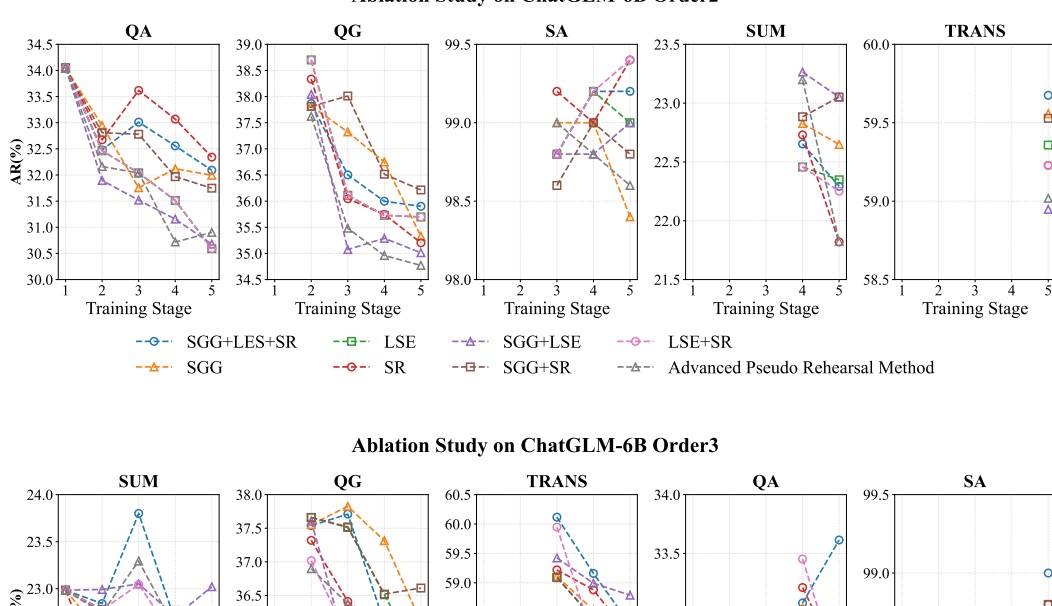

Figure 9: Ablation results detailing the performance variations of different models across different task chains

# E   Real-Sample Rehearsal Details

This section presents additional results on real-sample rehearsal for comparison with pseudo-sample rehearsal. As shown in Table E, even when 10% of real samples are used for rehearsal under the same continual learning setup, the performance remains lower than that of pseudo-sample rehearsal. This observation can be explained by the fact that labels synthesized by the old model facilitate learning, improving the new model's task adaptation. In contrast, real-sample rehearsal is constrained by the limited number of available samples, resulting in reduced diversity and weaker knowledge coverage. Consequently, its performance degrades more noticeably under low rehearsal ratios.

Table 5: Real-Sample Rehearsal Results on LLaMA2-7B

| Data Rehearsal | Order1 | Order2 | Order3 | Avg |
|---|---|---|---|---|
| 1% real samples | 48.11 | 49.02 | 48.74 | 48.62 |
| 5% real samples | 50.18 | 50.65 | 50.02 | 50.28 |
| 10% real samples | 50.24 | 51.09 | 50.84 | 50.73 |
| 1% real samples synthesis 10% pseudo samples | 52.90 | 53.01 | 52.84 | 52.92 |

# F   Comparison Study Details

In this section, we provide additional experimental details on the ChatGLM-6B model to demonstrate the impact of the hyperparameters $k$ and $\alpha$ on the SERS framework.

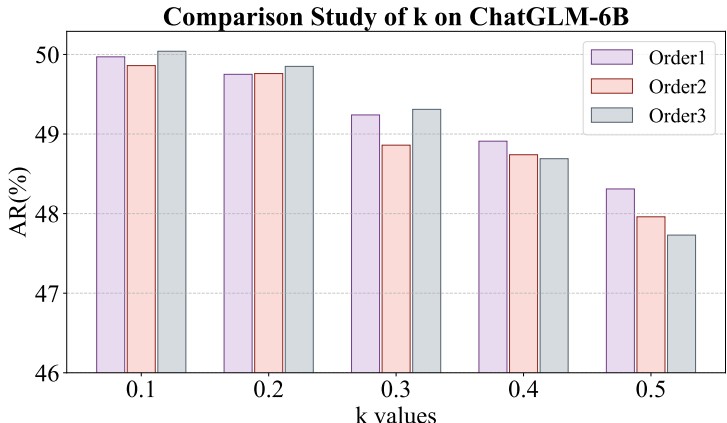

Figure 10: Comparison study of $k$ ($\alpha$=0.6) on ChatGLM-6B

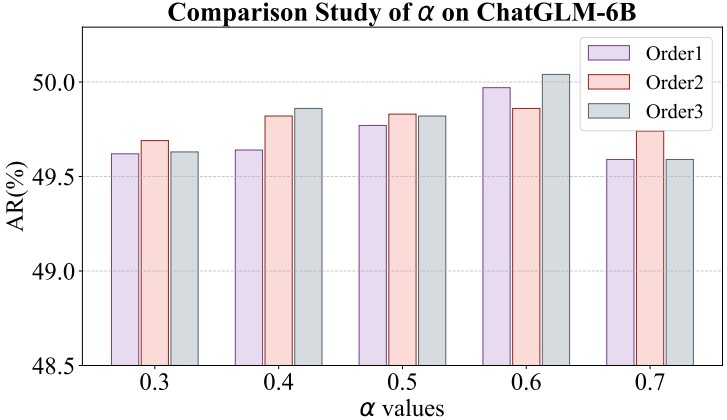

Figure 11: Comparison study of $\alpha$ ($k$=0.1) on ChatGLM-6B

