# OpenReview forum: "Self-Evolving Pseudo-Rehearsal for Catastrophic Forgetting with Task Similarity in LLMs"
_NeurIPS.cc/2025/Conference — NeurIPS 2025 poster_

### Official Review · Reviewer_36jW · 2025-06-29

**Clarity:** 4
**Significance:** 3
**Originality:** 3
**Rating:** 4
**Confidence:** 4

**Summary:**

This paper proposes Self-Evolving Pseudo-Rehearsal with Task Similarity (SERS), a continual learning framework for large language models (LLMs) for addressing catastrophic forgetting. SERS decouples pseudo-input synthesis from label creation - generating pseudo inputs via template guidance and semantic masking and uses a label self-evolution module to prevent over-specialization in pseudo labels.
Moreover, SERS introduces a dynamic regularizer driven by the Wasserstein distance between task distributions to balance stability and plasticity. Experiments on different LLMs demonstrate consistent improvements over several rehearsal, regularization, and structure-based baselines, achieving relative gains in Rouge-L and backward transfer metrics.

**Questions:**

* Is continual learning LLM more of conceptual value but limited current production adoption? Likely using prompt engineer or API models with no internal training is more popular.
* Does SERS work well with efficient tuning methods, e.g. LORA and adapters?

**Ethical Concerns:**

["NO or VERY MINOR ethics concerns only"]

**Final Justification:**

I keep my current rating as the authors addressed my questions regarding continual learning LLM vs prompt tuning, as well as applying SERS to LORA and adapters.

**Limitations:**

Yes

**Quality:**

3

**Strengths And Weaknesses:**

* Strengths
  * This paper is well motivated and easy to follow.
  * The weaknesses in existing replay, pseudo-rehearsal and self-synthesis methods for continual LLM tuning paper are clearly identified, and novelty of this paper over prior methods is well illustrated.
  * The proposed methodology is clearly described and well-justified. Comprehensive experiments across multiple tasks and models (LLaMA2-7B, ChatGLM-6B) demonstrate the method’s superiority over SOTA methods (SSR, SAPT from 2024).

* Weaknesses
  * It is helpful to show qualitative analysis or visualization of the similarity metric.
  * It is helpful to show qualitative analysis or visualization of the pseudo-samples.

---

> ### Author Rebuttal · Authors · 2025-07-30
>
> **To Reviewer 36jW:**
>
> We sincerely appreciate for your constructive and valuable comments, which will play an important role in improving the paper. We hope that our responses will help you better understand our work.
>
> ***Q1: Further Analysis on Task Similarity Measurement***
>
> We apologize for the limited discussion on similarity measurement due to space constraints. From a qualitative perspective, the Wasserstein (WS) distance measures the difference between two probability distributions, defined as the minimum "transportation cost" required to move one distribution to another. Liu et al.[1] have provided theoretical analysis demonstrating that this method can be effectively used for quantifying task similarity. Wasserstein GANs[2], for instance, utilize this distance to assess the similarity between generated data and real data in the adversarial learning domain.
>
> In the context of continual learning, task "similarity" typically refers to how closely their feature distributions align in the feature space. **If the feature distributions of new and old tasks are very similar, the model is less likely to experience catastrophic forgetting while learning the new task. On the other hand, if the feature distributions differ significantly, the model must apply stronger parameter constraints to mitigate forgetting.** This concept forms the basis of our similarity regularization approach.
>
> ***Q2: Further Analysis on Pseudo-Sample Generation***
>
> We hope that Figures 1 and Figure 7 in Appendix A help address your concern regarding the visualization of pseudo-sample generation. As shown in Figure 1, **compared to the SSR[3] method (blue), our generated pseudo samples (green) exhibit better similarity with real samples (red) and also demonstrate improved diversity, unlike SSR's more concentrated pseudo samples.** In Figure 7, we present real data from the dataset to further illustrate the differences between SSR generated and our generated pseudo samples. It is evident that the SSR approach fails to capture the essence of the "cosmosqa" task, producing generic QA samples that lack key common-sense reasoning cues. In contrast, our method generates pseudo samples that closely resemble real ones at low masking ratios, and even with higher masking, the outputs remain diverse while retaining the core features of the "cosmosqa" task.
>
> ***Q3: Applications of Continual Learning and Comparison with Prompt and API Methods***
>
> 1.The Widespread Demand for Continual Learning
>
> As the scale of LLMs continues to grow, the data volume is also rapidly increasing, making it impractical to retrain the model using all data each time. On the other hand, fine-tuning only with new data can lead to catastrophic forgetting, where the model forgets previously learned knowledge. Continual learning methods exhibit significant value in addressing these challenges.
>
> 2.Application of Prompt and API Methods in Continual Learning
>
> Your observation is astute, and indeed, prompt methods are commonly used in continual learning. However, **since prompt methods require fewer trainable parameters and typically freeze the LLM's core components, they are usually applied in class-incremental learning and domain-incremental learning scenarios with relatively small knowledge spans.** In contrast, task-incremental learning, which is the focus of our work, involves a greater knowledge gap between tasks, making it challenging for the lightweight, trainable parameters of prompt methods to effectively learn multi-task knowledge. As a result, most advanced task-incremental continual learning methods rely on replay, architecture, and regularization strategies.
>
> **API methods address catastrophic forgetting from a different perspective.** While they have certain advantages, they also pose challenges, notably the need to access an external API during each inference. This introduces additional latency and computational overhead. In comparison, continual learning methods offer a more efficient, cost-effective alternative by enabling the model to learn new tasks while retaining old knowledge. **By continuously training and fine-tuning the model locally, continual learning avoids the reliance on external APIs, thus reducing computational and time costs associated with external access,which remains a popular approach to address catastrophic forgetting.**
>
> ***Q4: Can SERS be effectively combined with efficient fine-tuning methods such as LoRA and adapters?***
>
> **SERS is fully compatible with the LoRA method.** For the LLaMA2-7B model, we apply LoRA to the q_proj and v_proj layers; for the ChatGLM-6B model, we apply LoRA to the query_key_value layers. This approach aligns with advanced pseudo-sample replay methods such as SSR.
>
> As for the adapter method, although we employ LoRA for PEFT in SERS rather than the adapter method, **we do not make architectural changes to the model. Therefore, even with the use of adapter methods, we believe they can be effectively combined with SERS.** We will explore and incorporate methods that combine SERS with adapter techniques to provide additional PEFT options.
>
> We would like to express our sincere gratitude once again for your insightful comments, and we hope the above responses address your concerns. We will attend to all comments to the best extent in the revised version.
>
> *[1] Liu, Xinran, et al. "Wasserstein task embedding for measuring task similarities." Neural Networks 181 (2025): 106796.*
>
> *[2 ]Arjovsky, Martin, Soumith Chintala, and Léon Bottou. "Wasserstein generative adversarial networks." International conference on machine learning. PMLR, 2017.*
>
> *[3] Huang, Jianheng, et al. "Mitigating Catastrophic Forgetting in Large Language Models with Self-Synthesized Rehearsal." Proceedings of the 62nd Annual Meeting of the Association for Computational Linguistics (Volume 1: Long Papers). 2024.*

---

> > ### Comment · Reviewer_36jW · 2025-08-03
> >
> > Thank you for the response and for addressing the concerns raised in my review.

---

### Official Review · Reviewer_fHFB · 2025-06-30

**Clarity:** 4
**Significance:** 3
**Originality:** 3
**Rating:** 5
**Confidence:** 3

**Summary:**

The work introduces a novel method for fine-tuning LLMs within the continual learning setup. The method is a combination of a pseudo-rehearsal and regularization strategies. The evaluation of the method shows a clear advantage of the method over the baselines and closes the gap to the Multi-Task reference, which is a sign of a really strong performance.

**Questions:**

I've listed most of my questions in the Strengths & Weaknesses section.

**Ethical Concerns:**

["NO or VERY MINOR ethics concerns only"]

**Final Justification:**

Given my positive rating since the beginning and the thoughtful answers provided by the authors during the rebutal period, I keep my recommendation to accept the article.

**Limitations:**

Yes.

**Quality:**

3

**Strengths And Weaknesses:**

### Strengths
1. The paper is written clearly, and all the concepts, motivations behind the whole work, and respective experiments are well articulated.
2. The introduced method is correctly evaluated using a metric typical for continual learning. In both metrics (AR, BWT), the method beats other baselines, showing strong performance. Additionally, the authors took care of the order of tasks, which is known to be an important factor in CL experiments.
3. The idea to create Semantic guided Pseudo-Input and Self-Evolving Labels is interesting, and through ablations and examples, the authors highlight the role of these components in the overall performance.
4. The method requires storing as little as 1\% of the overall data samples to use them as a "skeleton" for artificial sample creation. This is a nice idea to reduce the requirements for the storage of samples. I'm wondering how far this process can go. I.e., I'd like to see the performance degradation of the overall method when further reducing the number of samples stored in the buffer. These experiments would nicely complement the findings from the experiment in Figure 5.
### Weaknesses
1. The length of the take sequence is relatively short. While I'm aware that it's harder to create equally long sequences of tasks as in the case of computer vision datasets, we know that longer sequences have a different set of challenges; thus, it'd be beneficial to see how this method works on such tasks.
2. There is little elaboration on why Wasserstein Distance is the right way to measure task similarity. While I like the idea of adjusting the level of regularization based on the task similarity, I find it quite disappointing that this (quite important!) choice is not discussed (or compared) in a broader context.
3. The proposed method for generating pseudo-samples seems to be particularly suited for the specific QA type of datasets. Do you have any ideas how this method could be adjusted to make it applicable to other types of datasets? Have you performed any experiments with different datasets?

---

> ### Author Rebuttal · Authors · 2025-07-30
>
> **To Reviewer fHFB:**
>
> We sincerely appreciate for your constructive and valuable comments, which will play an important role in improving the paper. We hope that the additional experiments and responses will help you better understand our work.
>
> ***Q1: Effectiveness with Fewer Real Samples***
>
> Following your suggestion, **we conducted experiments using fewer real samples to generate pseudo-samples.** Given that the benchmark training set contains 2,000 samples, 1% corresponds to just 20 samples. Since our pseudo-sample generation method relies on combinations of real samples, an extremely small number of real samples will lead to repetition. Therefore, we tested with 15 (0.75%) and 10 (0.5%) real samples, with results shown below.
>
> **The results show that 0.75% and 0.5% real samples slightly outperform 1% real samples when synthesizing 5% pseudo samples. However, in the remaining cases, they are less effective than synthesizing with 1% real samples.** This is particularly evident when a large number of pseudo-samples are generated, where performance drops significantly. This observation aligns with the conclusion presented in our paper. Since real samples are obtained through k-means clustering, **fewer real samples better capture core knowledge when synthesizing a small number of pseudo samples. In contrast, when synthesizing a larger number of pseudo-samples, the repetition of generated samples becomes too strong.**
>
> | Real Samples | Synthesized Pseudo Samples | Trans | SA | QA | SUM | QG | AR Metric |
> | --- | --- | --- | --- | --- | --- | --- | --- |
> | 15 (0.75%) | 100 (5%) | 62.35 | 98.8 | 35.02 | 26.51 | 39.57 | 52.45 |
> | 15 (0.75%) | 200 (10%) | 63.95 | 98.8 | 35.77 | 26.45 | 39.49 | 52.89 |
> | 15 (0.75%) | 300 (15%) | 64.17 | 98.8 | 35.74 | 25.89 | 38.98 | 52.71 |
> | 15 (0.75%) | 400 (20%) | 64.49 | 98.8 | 35.21 | 25.95 | 37.97 | 52.48 |
> | 10 (0.5%) | 100 (5%) | 62.78 | 98.8 | 34.11 | 25.83 | 39.64 | 52.23 |
> | 10 (0.5%) | 200 (10%) | 64.00 | 98.8 | 35.46 | 26.46 | 39.23 | 52.79 |
> | 10 (0.5%) | 300 (15%) | 64.09 | 98.6 | 35.26 | 25.40 | 38.63 | 52.40 |
> | 10 (0.5%) | 400 (20%) | 64.01 | 98.4 | 34.18 | 26.18 | 38.78 | 52.31 |
>
> **Q2: Short Task Sequence Length**
>
> In our experiments, **considering the size of the LLM parameters, we followed the task sequence length used in SSR[1] and conducted experiments with 5-task and 10-task sequences,** as shown in Table 1. The detailed task selection and sequence can be found in Table 3 and Table 4 of Appendix C. **Compared to other previous works in continual learning, the sequence length is representative in the context of task-incremental continual learning.** Extending the task sequence further would introduce more complex challenges. Some architecture-based methods such as SAPT[2] require expanding modules for each new task, while **our approach can effectively balance stability and plasticity without continuously increasing the training parameters, allowing it to still perform relatively well in long sequence tasks.**
>
> ***Q3: Theoretical Explanation of Task Similarity Regularization***
>
> In SERS, we use Wasserstein (WS) distance to quantify task similarity. **The WS distance measures the difference between two probability distributions, defined as the minimum "transportation cost"** to move one distribution to another. In other fields, **WS distance has been employed for similarity quantification, such as in Wasserstein GANs[3]**, where it compares generated data to real data.
>
> The reason it is effective in our work for quantifying task similarity is that, **in the context of continual learning, task "similarity" typically refers to how close their feature distributions are in the feature space.** If the feature distributions of the new and old tasks are very similar, the model is less likely to face catastrophic forgetting when learning the new task. On the other hand, if the feature distributions differ significantly, the model needs to apply stronger parameter constraints to mitigate forgetting. This is the core idea behind our similarity regularization.
>
> ***Q4: Is the Pseudo-Sample Generation Specific to QA Tasks?***
>
> For ease of visualization, we used a simple QA task in the figures of the paper. However, **our pseudo-sample generation method is generalizable to text-to-text tasks. Our dataset also includes other tasks such as question generation (QG), sentiment analysis (SA), translation (Trans), and summarization (SUM)**. As shown in the light yellow line in Figure 4, which optimizes only the pseudo-sample generation process, our method can generate high-quality pseudo-samples for these tasks.
>
> Due to the generality of our masking strategy, we believe our method could also generalize to multimodal tasks. We plan to further explore this potential in future work.
>
> We would like to express our sincere gratitude once again for your insightful comments, and we hope the above responses address your concerns. We will attend to all comments to the best extent in the revised version.
>
> *[1] Huang, Jianheng, et al. "Mitigating Catastrophic Forgetting in Large Language Models with Self-Synthesized Rehearsal." Proceedings of the 62nd Annual Meeting of the Association for Computational Linguistics (Volume 1: Long Papers). 2024.*
>
> *[2]Zhao, Weixiang, et al. "SAPT: A Shared Attention Framework for Parameter-Efficient Continual Learning of Large Language Models." Proceedings of the 62nd Annual Meeting of the Association for Computational Linguistics (Volume 1: Long Papers). 2024.*
>
> *[3] Arjovsky, Martin, Soumith Chintala, and Léon Bottou. "Wasserstein generative adversarial networks." International conference on machine learning. PMLR, 2017.*

---

> > ### Comment · Reviewer_fHFB · 2025-08-07
> >
> > I would like to thank the authors for a detailed response clarifying my doubts. While I keep my score the same, I find this work to be insightful and work-accepting, thus I will vote for clear acceptance.

---

### Official Review · Reviewer_izSP · 2025-07-01

**Clarity:** 3
**Significance:** 2
**Originality:** 3
**Rating:** 4
**Confidence:** 3

**Summary:**

This paper presents Self-Evolving Pseudo-Rehearsal with Task Similarity (SERS), a novel continual learning framework designed to mitigate catastrophic forgetting in large language models (LLMs). SERS introduces three key innovations: (1) it decouples pseudo-input generation from label synthesis by leveraging semantic masking and template-guided prompt construction; (2) it incorporates a self-evolution mechanism that adaptively fuses base and task-specific label signals to improve generalization; and (3) it employs a task-similarity-aware regularizer based on the Wasserstein distance to dynamically control the stability–plasticity trade-off. Evaluations on the SuperNI benchmark using LLaMA2-7B and ChatGLM-6B across diverse task permutations demonstrate that SERS consistently outperforms state-of-the-art pseudo-rehearsal and regularization methods, and in some cases achieves performance comparable to or better than multi-task training.

**Questions:**

see above

**Ethical Concerns:**

["NO or VERY MINOR ethics concerns only"]

**Final Justification:**

This is a borderline paper. Paper is written well and easy to follow. However, the proposed method is sensitive to the order of tasks, which limits it applications. I choose borderline accept.

**Limitations:**

see above

**Quality:**

3

**Strengths And Weaknesses:**

Strengths:
1. The SERS framework integrates pseudo-rehearsal with dynamic, task-similarity-aware regularization, jointly addressing the stability–plasticity dilemma and the fidelity of pseudo-samples. Its use of semantic masking and template-guided input synthesis is well-motivated and yields pseudo-data that closely aligns with the true task distribution.
2. The proposed two-stage mechanism, which consists of a self-questioning phase followed by the integration of outputs from both pre-trained and task-adapted models, aligns well with recent developments in self-evolution frameworks. This design effectively mitigates over-specialization by maintaining a balance between generalization and task-specific adaptation.

Weakness:
1. While the proposed method is effective in benchmark settings, its applicability to real-world domain-specific scenarios, such as medicine or law,  is not well explored. In such cases, large language models already perform well on general tasks, and continual learning becomes necessary only when adapting to specialized domains. However, determining which domain-specific knowledge should be retained and what type of pseudo-samples should be synthesized remains an open and non-trivial challenge.
2. The framework relies on measuring task similarity to guide regularization, but the method for quantifying meaningful relationships between heterogeneous tasks is insufficiently justified. For example, assessing similarity between a mathematical reasoning task and a clinical diagnosis task poses significant challenges due to their fundamentally different semantics. Although the experiments are conducted on the SuperNI benchmark, the task diversity in this dataset may not fully capture the complexity of real-world task heterogeneity.

---

> ### Author Rebuttal · Authors · 2025-07-30
>
> **Response to Reviewer izSP:**
>
> We sincerely appreciate for your constructive and valuable comments, which will play an important role in improving the paper. We hope that our responses will help you better understand our work.
>
> ***Q1: Applicability in Real-World Domain-Specific Scenarios, Knowledge Retention, and Pseudo-Sample Generation***
>
> To enable comparison with state-of-the-art methods, we conducted our study using the widely adopted SuperNI benchmark in continual learning. Nevertheless, we believe the SERS method is also applicable to real-world, domain-specific tasks. For example, **in the legal domain, tasks like Question Answering, Summarization, and Text Classification in the multilingual dataset LawInstruct[1]  share similarities with our dataset.** Similarly, **in other domains where the model faces the challenge of learning new tasks with new data while retaining the ability to preserve old tasks, we believe SERS will also exhibit strong generalization capability.**
>
> For knowledge retention: Knowledge retention strategies are more evident in architecture-based continual learning methods, which add extra parameter blocks to store task-specific knowledge. However, this approach requires continuously expanding the parameter space, making it difficult to accommodate long task sequences. In contrast, SERS employs a rehearsal strategy, allowing the model to balance plasticity and stability by reviewing old knowledge while learning new tasks.
>
> For pseudo-sample generation: SERS leverages the in-context learning of large models. By using a few real samples, SERS generates pseudo-samples for each task that effectively reflect domain-specific knowledge, helping the model review old knowledge during the training of new tasks.
>
> ***Q2: Task Similarity Quantification***
>
> We appreciate your thoughtful feedback. In SERS, we use the Wasserstein (WS) distance to measure task similarity. **Liu et al.[2] have theoretically proven the effectiveness of this quantification method.** This similarity is applied through a regularization loss in continual learning tasks, which quantifies the **relative differences between new and old tasks.** As you mentioned, a mathematical reasoning task and a clinical diagnosis task have fundamental semantic differences, so it is indeed more likely for the model to experience forgetting when learning a clinical diagnosis task after a mathematical reasoning task. Therefore, a larger regularization loss constrains parameter changes to mitigate forgetting. In contrast, when the model learning a statistical analysis task after a mathematical reasoning task, a smaller regularization loss reduces the constraint and encourages knowledge transfer.
>
> ***Q3: Dataset Diversity and Its Gap with Real-World Tasks***
>
> We completely understand your concern; the diversity of the dataset and its alignment with real-world tasks are indeed crucial factors in determining the practical applicability of our method. **The SuperNI benchmark encompasses a variety of classic text-to-text tasks**, such as question answering (QA), question generation (QG), sentiment analysis (SA), summarization (SUM), and translation (TRANS), all of which exhibit a good degree of diversity. **Moreover, the selected sub-tasks in our study are closely related to real-world tasks.** For example, QA comes from real conversations on large social platforms, QG includes questions extracted from Wikipedia articles, SA is based on product reviews from Amazon, SUM is from Reddit’s "Today I Fucked Up" (TIFU) section, and Trans is derived from TED Talks. All of these tasks are based on common, real-life scenarios and represent familiar question types. We will also further experiment with domain-specific real-world datasets such as LawInstruct and supplement the results in the final version.
>
> We would like to express our sincere gratitude again for your insightful comments, and we hope the above responses address your concerns. We will attend to all comments to the best extent in the revised version.
>
> *[1] Niklaus, Joel, et al. "LawInstruct: A Resource for Studying Language Model Adaptation to the Legal Domain." Findings of the Association for Computational Linguistics: NAACL 2025. 2025.*
>
> *[2] Liu, Xinran, et al. "Wasserstein task embedding for measuring task similarities." Neural Networks 181 (2025): 106796.*

---

> > ### Comment · Reviewer_izSP · 2025-08-05
> > **Response to authors**
> >
> > Thank you for your response. I do not have concerns about the specific metric used to measure task similarity. My main concerns are: 1) Whether the proposed method is sensitive to the order in which tasks are presented, and 2) If such sensitivity exists, how the method is designed to address or mitigate this issue.
> >
> > I also admit it is a non-trivial problem to address it. I keep my score as Borderline accept.

---

> ### Author Response · Authors · 2025-08-07
> **Response to reviewer izSP**
>
> We sincerely appreciate your  valuable comment. We hope that our response provides a clearer explanation of the design of our method regarding task sequence robustness and effectively addresses your concerns.
>
> **Q1: Task Sequence Robustness and Similarity Regularization**
>
> Your comment is insightful. The robustness of continual learning methods to the sequence in which tasks are presented is indeed a point worth discussing, and we have made corresponding improvements in our work. Traditional rehearsal methods often struggle to maintain consistent performance across different task sequences. In contrast, one of the key motivations behind our task similarity regularization is to enhance the robustness of continual learning against variations in task order. Intuitively, without similarity regularization, the model is more prone to catastrophic forgetting when task sequences differ significantly. However, by incorporating similarity regularization, the model imposes stronger parameter constraints when tasks differ substantially, thereby alleviating forgetting. From the experimental results, it is evident that advanced methods such as SSR exhibit significant performance variability across three different task sequences, whereas our method demonstrates stronger robustness, further validating our approach. Thus, our method shows improved task sequence robustness compared to other rehearsal methods.
>
> We truly value your comments and hope that our response provides a clear explanation of the issues you raised.

---

### Official Review · Reviewer_ZMGj · 2025-07-03

**Clarity:** 2
**Significance:** 2
**Originality:** 3
**Rating:** 4
**Confidence:** 3

**Summary:**

This paper addresses a problem of pseudo sample replay in continual learning, where the generated pseudo samples cannot well reflect the original knowledge structure and thus provide limited support for rehearsal. It proposes a method that first generates pseudo samples with semantic guidance and masked inputs, then performs label self-evolution by mixing the outputs of the current and initialized LLM. It also adds a regularization term to encourage more knowledge transfer across similar tasks. Results show that the proposed method outperforms other baselines.

**Questions:**

What is the performance gap between replay with pseudo examples and real examples? In the rehearsal analysis in Fig 5, using 1% real data can generate better pseudo samples for rehearsal than using 5%. Does using 1% real data also outperforms 5% real data in rehearsal?

**Ethical Concerns:**

["NO or VERY MINOR ethics concerns only"]

**Final Justification:**

The authors' rebuttal have addressed most of my concerns: the efficacy of pseudo samples and the influence of masking. But as mentioned in the rebuttal, if the tasks are dissimilar, the method will sacrifice plasticity to reduce forgetting which I believe can be further improved. So I raised my score to 4.

**Limitations:**

The model needs extra LLM forwarding to generate pseudo examples and label evolution. It also needs extra memory to store the LLM_0 and LLM_0, which is natural for PEFT models but have extra cost for fine-tuning models.

**Paper Formatting Concerns:**

No.

**Quality:**

2

**Strengths And Weaknesses:**

**Strengths**
1. The paper studies an important pseudo rehearsal problem for CL with LLM.
2. The experiments are performed on two LLMs, with thorough comparison to baselines. Ablation studies are included.

**Weaknesses**
1. Some details are unclear in the method:
* When masking the real sample for pseudo sample generation, how to decide the tokens to mask? Is it random masking? If so, the masked place may not contain much semantic information.
* How to integrate the output from different models? Is it combining a sequence of output tokens or representations? What are the real output tokens after integration?

2. The intuition for task similarity regularization is unclear to me. During pseudo example generation and label evolution, the model already preserves the information of LLM_0 for all tasks. Then why should it take additional regularization for task similarity? If the tasks in the CL sequence are dissimilar, will it limit the model’s plasticity to learn each task?

---

> ### Author Rebuttal · Authors · 2025-07-30
>
> **Response to Reviewer ZMGj:**
>
> We sincerely appreciate for your constructive and valuable comments, which will play an important role in improving the paper. We hope that the additional experiments and responses will help you better understand our work.
>
> ***Q1: Masking Strategy and Semantic Information***
>
> This is a good comment. As you have correctly noted, we indeed use a random masking approach. **Random masking has been widely used in related generation tasks. For example, Huang et al.[1] employed random masking to generate pseudo anomalies in unsupervised video anomaly detection, and we have adopted this approach as well.** We have considered the semantic information in the masked parts, where a larger dynamic masking ratio and post-processing clustering help preserve the semantic content. **Our masking ratio, set between 20% and 60%, which allows for better coverage of semantic content**, while filtering and clustering after pseudo-sample generation help improve the quality and diversity. The following real-world example may help you intuitively experience the effect of our masking.
>
> *Before Masking*: The item was said to have been delivered Friday and was not delivered. The shipping address was correct and payment information correct but the tracking information was wrong or it was never actually shipped.
>
> *After Masking*: The ____ said to have been ____ and was not delivered. The ____ was ____ and ____ information correct but ____ was wrong or it was never ____ shipped.
>
> ***Q2: Model Output Merging***
>
> To integrate the outputs from the current model and the initial model, **we weight their logits using the weight parameter α and select the token with the highest probability as the final output.** By leveraging the general knowledge from the initial model, we smooth the specialized knowledge of the current model, avoiding biases caused by over-specialized pseudo labels during the rehearsal process.
>
> ***Q3: Intuition and Necessity of Similarity Regularization***
>
> Thank you for pointing out this issue. Pseudo-sample generation is closely tied to task similarity regularization. **The goal of pseudo-sample generation and label evolution is to represent old task knowledge as data. Task similarity regularization governs how this knowledge is revisited and balances old knowledge retention with new task learning.**  Intuitively, it imposes constraints on parameter changes based on task similarity. When tasks are similar, regularization is relaxed, facilitating knowledge transfer. If tasks differ significantly, regularization becomes stricter to prevent forgetting key old task knowledge.
>
> If tasks in the CL sequence are dissimilar, similarity regularization prioritizes stability over plasticity, sacrificing some plasticity to mitigate catastrophic forgetting. This is the core motivation behind our similarity regularization design.
>
> ***Q4: Comparison between Pseudo-Sample Rehearsal and Real-Sample Rehearsal***
>
> This is an insightful and practically valuable suggestion. Regarding Figure 5, when synthesized pseudo samples are few (≤15%), 1% real samples outperforms 5% real samples. Since real samples are selected through clustering, when fewer pseudo samples are synthesized, 1% real samples lead to fewer clustering centers, resulting in pseudo samples that better reflect core knowledge. However, with a higher count of synthesized pseudo samples (20%), 5% real samples provide more comprehensive knowledge, approaching multi-task learning performance.
>
> Following your suggestion, **we included real-sample rehearsal results, as shown in the table below.** **Our findings align with those of the SSR[2] method, where even with 10% real sample rehearsal, the performance is lower than using pseudo samples.**  This is because labels synthesized by the old model facilitate learning, improving the new model’s task adaptation. However, unlike pseudo-sample rehearsal, where different numbers of real samples are used to synthesize the same quantity of pseudo samples, **the 1% real sample rehearsal did not achieve the same results as the 5% real sample rehearsal due to the smaller rehearsal ratio**.
>
> | Data Rehearsal | Order 1 | Order 2 | Order 3 | Avg |
> | --- | --- | --- | --- | --- |
> | 1% real samples | 48.11 | 49.02 | 48.74 | 48.62 |
> | 5% real samples | 50.18 | 50.65 | 50.02 | 50.28 |
> | 10% real samples | 50.24 | 51.09 | 50.84 | 50.73 |
> | 1% real samples synthesis 10% pseudo samples | 52.90 | 53.01 | 52.84 | 52.92 |
>
> ***Q5: The Need for Storing LLM_0***
>
> We fully understand your concern. Storing LLM_0 dose imposes some burden, and storage is one of the challenges faced by pseudo-sample rehearsal methods. However, it is worth noting that most pseudo-sample rehearsal methods also rely on additional generative modules or LLMs for pseudo-sample generation. For instance, the SAPT[3] method depends on training an additional generative module, which increases the training burden, while the SSR method similarly uses LLM_0 for generating pseudo samples. **Although our work requires storing LLM_0, it only involves forward inference, independent of the training process, and does not introduce additional training costs.** We will also further explore more lightweight storage solutions.
>
> We would like to express our sincere gratitude again for your insightful comments, and we hope the above responses address your concerns. We will attend to all comments to the best extent in the revised version.
>
> *[1] Huang, Xiangyu, et al. "Synthetic pseudo anomalies for unsupervised video anomaly detection: A simple yet efficient framework based on masked autoencoder." ICASSP 2023-2023 IEEE International Conference on Acoustics, Speech and Signal Processing (ICASSP). IEEE, 2023.*
>
> *[2] Huang, Jianheng, et al. "Mitigating Catastrophic Forgetting in Large Language Models with Self-Synthesized Rehearsal." Proceedings of the 62nd Annual Meeting of the Association for Computational Linguistics (Volume 1: Long Papers). 2024.*
>
> *[3] Zhao, Weixiang, et al. "SAPT: A Shared Attention Framework for Parameter-Efficient Continual Learning of Large Language Models." Proceedings of the 62nd Annual Meeting of the Association for Computational Linguistics (Volume 1: Long Papers). 2024.*

---

> > ### Comment · Reviewer_ZMGj · 2025-08-04
> >
> > Thank the authors for the detailed explanation and clarification. I have raised my score to 4.

---

### Note · Authors · 2025-08-12

Dear Reviewers,

Thank you for your valuable review. We have provided responses to your questions and have made efforts to address the concerns.
Specifically, we have elaborated on the details of the pseudo-sample generation strategy, provided the theoretical foundation for task similarity regularization, and demonstrated the diversity of the dataset and the model's generalization capabilities. Additionally, we have supplemented experiments using direct real-sample rehearsal and fewer real samples to generate pseudo samples for rehearsal, further validating the superior performance of our method.

We truly appreciate the time and effort you have invested in reviewing our paper. We will attend to all comments to the best extent in the revised version.

Best regards,

Authors of Submission#9296

---

### Decision · Program_Chairs · 2025-09-17

**Decision:**

Accept (poster)

**Comment:**

This paper addresses catastrophic forgetting in LLM continual fine-tuning by introducing a pseudo-rehearsal framework that combines semantic masking, label self-evolution, and task-similarity regularization. The reviewers were generally positive finding the method motivated and strong experiments on multiple LLMs and diverse tasks, showing consistent improvements. While reviewers raised concerns about the justification of some design choices the main concerns were addressed in the rebuttal. Overall, the paper makes a technically solid and timely contribution